# Genome-Wide and Comprehensive Analysis of the Multiple Stress-Related CAF1 (CCR4-Associated Factor 1) Family and Its Expression in Poplar

**DOI:** 10.3390/plants10050981

**Published:** 2021-05-14

**Authors:** Pu Wang, Lingling Li, Hui Wei, Weibo Sun, Peijun Zhou, Sheng Zhu, Dawei Li, Qiang Zhuge

**Affiliations:** Co-Innovation Center for Sustainable Forestry in Southern China, Key Laboratory of Forest Genetics & Biotechnology, Ministry of Education, College of Biology and the Environment, Nanjing Forestry University, Nanjing 210037, China; wangpu@njfu.edu.cn (P.W.); linglingli@njfu.edu.cn (L.L.); HW@njfu.edu.cn (H.W.); czswb@njfu.edu.cn (W.S.); pjzhou@njfu.edu.cn (P.Z.); zhusheng303@163.com (S.Z.); dwli@njfu.edu.cn (D.L.)

**Keywords:** CCR4 associated factor 1, poplar, gene expression, multiple stresses

## Abstract

Poplar is one of the most widely used tree in afforestation projects. However, it is susceptible to abiotic and biotic stress. CCR4-associated factor 1 (*CAF1*) is a major member of CCR4-NOT, and it is mainly involved in transcriptional regulation and mRNA degradation in eukaryotes. However, there are no studies on the molecular phylogeny and expression of the *CAF1* gene in poplar. In this study, a total of 19 *PtCAF1* genes were identified in the *Populus trichocarpa* genome. Phylogenetic analysis of the *PtCAF1* gene family was performed with two closely related species (*Arabidopsis thaliana* and *Oryza sativa*) to investigate the evolution of the *PtCAF1* gene. The tissue expression of the *PtCAF1* gene showed that 19 *PtCAF1* genes were present in different tissues of poplar. Additionally, the analysis of the expression of the *PtCAF1* gene showed that the *CAF1* family was up-regulated to various degrees under biotic and abiotic stresses and participated in the poplar stress response. The results of our study provide a deeper understanding of the structure and function of the *PtCAF1* gene and may contribute to our understanding of the molecular basis of stress tolerance in poplar.

## 1. Introduction

Gene expression is an important factor in plant growth and environmental communication. The steady-state level of intracellular mRNA is determined by the rate of transcription and post-transcriptional regulation of mRNA decay [1,2]. Genome-wide studies show that gene expression undergoes extensive reprogramming under environmental stimulation, and the change in the mRNA degradation rate provides a mechanism for rapidly changing mRNA abundance [3,4]. Degradation of poly(A) tail is a key step in reducing and controlling the gene expression of messenger RNA (mRNA) [5,6,7]. The carbon decomposing metabolite inhibitor 4 (CCR4)-associated factor 1 in CCR4-NOT deaminase complex plays an important role in the process of mRNA deamination in most eukaryotes [8,9]. CCR4-NOT is a multisubunit protein complex, which is highly conserved in eukaryotes. It is mainly involved in transcription regulation, mRNA degradation, histone modification, and other important physiological processes [10,11,12]. *CAF1* (CCR4-associated factor 1) is also called CCR4-NOT transcription complex subunit 7 (CNOT7), which is highly conserved and mainly involved in transcriptional regulation and mRNA degradation [13,14,15]. *CAF1* is a highly conserved exonuclease in evolution. There is only a single gene in yeast, *Caenorhabditis Elegans,* and *Drosophilid* species. *CAF1* belongs to the DEDD family and is one of the major degradation enzymes involved in mRNA degradation [16,17,18]. Although *CAF1* has been widely studied in yeast and animals, its role has rarely been studied in plants.

Plants face a variety of abiotic and biotic stresses, such as plant pathogens, drought, cold, hot, and salt stress, in the process of growth, which are detrimental to their growth. However, plants have their own regulatory pathways with which to deal with various stresses [19]. Various studies have shown that *CAF1* plays an important role in plant growth, stress response, and resistance to microbial pathogens. *CAF1* is very important for the growth of yeast and animal cells, but they generally only have one or two *CAF1* homologous genes, unlike the higher plants, which contain the entire *CAF1* gene family [19,20,21,22]. Until now, 18 *CAF1* genes homologues have been identified in *Oryza sativa* and 11 *CAF1* genes in *Arabidopsis thaliana* [23]. Studies on the function of the *CAF1* gene in *A. thaliana* and *O. sativa* have been published. The results show that the *CAF1* gene plays an important role in plant growth and responses to biotic and abiotic stresses. Various studies have demonstrated that the overexpression of the pepper gene *CaCAF1* in tomato significantly promotes the growth of tomato and resistance against the oomycete pathogen *Phytophthora infestans* [24]. Moreover, *Citrus sinensis CAF1* (*CsCAF1*) is involved in the stress process of *Xanthomonas citri*. In addition, *CsCAF1* can interact with PTHA4, which is the main effective factor of *X. citri*. PTHA4 can stabilize the expression of CsLOB1 by inhibiting the deadenylase activity of *CsCAF1* and can make citrus resistant to canker [25]. In *A. thaliana*, the single and double mutants of *AtCAF1A* and *AtCAF1B* were shown to decrease the expression of PATHOGENESIS-RELATED 1 (PR1) and PATHOGENESIS-RELATED 2 (PR2), and plants become more susceptible to *Pseudomonas syringae pv tomato DC3000* (PST DC3000) infection; while plants overexpressing *AtCAF1A* showed high expression of PR1 and PR2 and exhibited enhanced resistance to the same pathogen [26]. The *CAF1* gene also plays an important role in abiotic stress response. In *A. thaliana*, abscisic acid (ABA), salicylic acid (SA), jasmonic acid (JA), methyl jasmonate (MeJA), and other hormones, as well as various external stresses, such as cold and wounds, can induce the rapid expression of the *CAF1* gene. Furthermore, using poly(A) tail length (PAT) analysis, it was confirmed that the *CAF1* gene has 3′-5′ amino acids. Exonuclease activity can regulate the degradation of mRNA in vivo. Stress tolerance experiments showed that *AtCAF1A* and *AtCAF1B* were involved in mediating abiotic stress responses [27,28]. In *O. sativa*, *OsCAF1B* plays an important role in the growth and development of *O. sativa* seedlings at low temperature. Moreover, it plays an important role in *O. sativa* germination and seedling growth [29,30]. In *Dunaliella salina*, the promoters of *DsCAF1* contain many elements that are sensitive to abiotic stress. *DsCAF1* was shown to be highly expressed in a high salt environment. When *D. salina* cells were subjected to stress shock, the expression of *DsCAF1* demonstrated a two-stage reaction. The mRNA level of *DsCAF1* increased 2–4-fold immediately after hypertonic, heat, or ultraviolet treatment, then increased 10-fold. In addition, it rapidly increased approximately 3-fold after hypotonic or cold shock, then decreased suddenly. The different expression patterns of *DsCAF1* indicate that *DsCAF1* is a nuclease in response to stress, which can regulate the mRNA expression of related genes [31].

Poplar (*Populus* spp.) is one of the most widely distributed and adaptable tree species in the world. Because of the value of poplar in the wood industry and its advantages for afforestation, it is important both economically and ecologically and plays an important role in the global ecosystem [32,33]. At present, the molecular mechanism of stress tolerance in poplar has not been fully elucidated. As a result of the availability of the *Populus trichocarpa* genome sequence, poplar has been used as a model species for the study of the perennial plant genome. At present, it is very important to improve poplar varieties by means of genetic engineering. However, there are no published papers concerning the *CAF1* gene in poplar. For this reason, it is necessary to study the *CAF1* gene family in poplar. In this study, we characterized the *CAF1* gene family in *P. trichocarpa* and analyzed its distribution, phylogenetic relationships, gene structure, and evolutionary characteristics. Moreover, we analyzed its expression in different tissues and its response to multiple environmental stresses. Our results provide insights that may of use for poplar breeding programs.

## 2. Results

### 2.1. Identification of the CAF1 Proteins in P. trichocarpa

To identify the *CAF1* genes in the genome of *P. trichocarpa*, we used the reported *CAF1* in *A. thaliana* and *O. sativa* as queries in Phytozome v12.1. All possible *PtCAF1* genes were excavated from the *P. trichocarpa* genome using the BLAST and HMMER search methods. After removing the redundant genes, 19 potential *PtCAF1* genes were screened out in the *P. trichocarpa* genome and named *PtCAF1A*-*PtCAF1S* based on their chromosomal location. The 19 potential *PtCAF1* genes were then identified and analyzed. The gene characteristics included the coding sequence length (CDS), protein amino acid quantity, protein molecular weight (MW), isoelectric point (PI), and subcellular localization (Table 1). The results showed that the 19 *PtCAF1* gene code sequence lengths ranged from 648 bp (*PtCAF1B*) to 1881 bp (*PtCAF1O*), and the 19 *PtCAF1* genes transcription factor proteins ranged from 216 AA to 627 AA; the protein sequence encoded by *PtCAF1O* was the longest (627 AA), and the protein sequence encoded by *PtCAF1B* was the shortest (216 AA). The MW of the proteins ranged from 24.78 KDa to 70.23 KDa, and the PI ranged from 4.41 (*PtCAF1Q*) to 9.91 (*PtCAF1C*). Understanding the subcellular localization of *PtCAF1* is very important for studying and evaluating gene function. The results showed that among the 19 *PtCAF1* genes, *PtCAF1A*, *PtCAF1B*, *PtCAF1C*, *PtCAF1D*, *PtCAF1F*, *PtCAF1I*, *PtCAF1J*, *PtCAF1L*, *PtCAF1M*, *PtCAF1P*, *PtCAF1Q*, *PtCAF1R*, and *PtCAF1S* are located in cytosol. *PCAF1E* and *PtCAF1G* are located in the nucleus. *PtCAF1H*, *PtCAF1K*, *PtCAF1N,* and *PtCAF1O* are located in the chloroplast. Most of the *PtCAF1* gene localization is in cytosol, which is consistent with its function.

### 2.2. Multiple Sequence Alignment and Phylogenetic Analysis

To explore the phylogenetic relationship of *PtCAF1* proteins in *P. trichocarpa*, we constructed a phylogenetic tree using the neighbor-joining (NJ) method based on multiple sequence alignments of *O. sativa*, *A. thaliana*, *Eucalyptus grandis*, *Elaeis guineensis* and *O. sativa*, *Capsicum annuum*, *Citrus sinensis*, *Nicotiana tabacum*, and *P. trichocarpa CAF1* protein sequences. According to previous studies on the *CAF1* gene family in *A. thaliana*, we divided the phylogenetic tree into three groups. The 19 candidate *PtCAF1* genes were divided into three groups: Group I, Group II, and Group III (Figure 1). To further evaluate the similar regions and conserved sites of the *PtCAF1* genes and clarify their evolutionary relationship among species, we used ClustalX 2.1 for multiple sequence alignment. The amino acid sequence analysis showed that the homologues of *A. thaliana*, *O. sativa*, *C. annuum*, *C. sinensis*, *N. tabacum*, yeast (*SpCAF1*), human (HsCNOT7), mouse, *Zea mays* (*ZmCAF1*), and *P. trichocarpa* (group I) were well conserved in the RNase D domain, with three conserved motifs and four important nuclease activity catalytic residues DEDD. In addition, these *CAF1* homologues contain the fifth conserved amino acid residue histidine (Appendix A).

### 2.3. Gene Structure and Motif Composition of PtCAF1 Gene Family

The exon/intron structure pattern of the *PtCAF1* genes and the conserved domain were studied according to their phylogenetic relationships. By comparing the genomic DNA sequences of the *PtCAF1* genes, we obtained their intron and exon structures. The motifs of the 19 *PtCAF1* genes were analyzed using online MEME software. According to the results of the MEME motif analysis, a schematic diagram was constructed to characterize the structure of the *PtCAF1* proteins. In order to elucidate this result, 10 conserved motifs were found in the *PtCAF1* proteins (Appendix A). Moreover, we constructed a phylogenetic tree using the *P. trichocarpa* protein sequence and *A. thaliana* protein sequence, which is consistent with the aforementioned phylogenetic tree. The coding sequence of the *PtCAF1* gene and the corresponding genomic DNA sequence were used to analyze the organization of exons/introns. The results are shown in Figure 2. There are differences in the number of introns between *CAF1* genes. Group III and Group I exhibit a similar number of introns, ranging from 0 to 1. However, in GroupII, *PtCAF1* has 0 to 7 introns. The analysis of the gene structure and phylogenetic tree showed that the *CAF1* gene between Group I and Group III experienced a series of evolutionary events leading to intron insertion, so they may have different functions. Group III genes contain eight motifs, and the types of motifs are the same. Group I genes contain similar motifs, with a maximum of eight motifs and a minimum of five motifs. Group II *PtCAF1* has seven motifs and two motifs. These results indicate that the most closely related members have similar exon/intron structures and similar protein motifs. By studying the 19 *PtCAF1* protein motifs, we found that the *PtCAF1* protein sequence was highly conserved, and 10 motifs in the 19 *PtCAF1* amino acid sequences were basically identical in the same family, especially in Group III. Studies of exon/intron structure have shown that most *PtCAF1* from the same subfamily have similar exon numbers. The motif composition and sequence of all genes were consistent, which was highly conserved in evolution.

### 2.4. Protein Structure Prediction of CAF1

There is a close relationship between the spatial structure and function of the protein, because the function of the protein is realized by changing its spatial conformation. In order to further determine the spatial structure of *CAF1*, we used the SWISS-MODEL website (https://swissmodel.expasy.org/ (accessed on 30 April 2021)) for homology modeling. All *PtCAF1* and *AtCAF1* proteins could be predicted as models, which indicates that they maintained their structural integrity during evolution, which plays an important role in their functions. In each branch, we selected one protein with the highest coverage from *P. trichocarpa*, *O. sativa,* and *A. thaliana*. Except for *PtCAF1R* (coverage 0.86) and *OsCAF1B* (coverage 0.8), the other confidence values are all higher than 90%. The results are shown in the Figure 3. We also found that the spatial conformations of the *P. trichocarpa*, *O. sativa,* and *A. thaliana* proteins belonging to the same branch are highly similar.

### 2.5. Promoter cis-Element Analysis

The 1.5 kb upstream sequences from *PtCAF1* were programed in the PlantCARE server (Appendix A). As shown in Appendix A, potential environmental factor-related cis-regulatory elements were predicted to be correlated with light and anaerobic induction response, which were most widely spread in the promoters of *PtCAF1*. The putative cis-acting regulatory DNA elements in *PtCAF1* were identified. In addition to the basic elements (CAAT-box and TATA-box), most *PtCAF1* genes had several cis-elements that were related to environmental stress signaling and phytohormone responses, such as LTR (low-temperature responsiveness), MBS (drought-inducibility), TATC-box (gibberellin-responsiveness), G-Box (light responsiveness), ABRE (the abscisic acid responsiveness), CGTCA-motif (the MeJA-responsiveness), ARE (anaerobic induction), MRE (light responsiveness), TC-rich repeats (defense and stress responsiveness), and TGACG-motif (the MeJA-responsiveness). It is well known that phytohormones, such as methyl jasmonate (MeJA) and abscisic acid (ABA), enable plants to adapt to abiotic stress. Thus, it is possible that most *PtCAF1* genes respond to abiotic stress.

### 2.6. Chromosomal Distribution and Synteny Analysis of PtCAF1 Genes and Ka/Ks Calculation

We identified 19 *PtCAF1* gene sequences in the *P. trichocarpa* genome using bioinformatics methods, while there are 11 *CAF1* genes in *A. thaliana* and 18 *CAF1* genes in *O. sativa*. This may be due to the specific expansion of the pedigree and the loss of copy number during the evolution of the *PtCAF1* gene family. Chromosome mapping of *PtCAF1* genes was performed using the *P. trichocarpa* genome database. A total of 19 *PtCAF1* genes were distributed on eight chromosomes. The largest number of *PtCAF1* genes was found on Chromosome 1 (five genes), while Chromosomes 9 and 16 had the smallest number of *PtCAF1* genes (one gene) (Figure 4). In addition, we analyzed the duplication events of *PtCAF1* genes in the *P. trichocarpa* genome (Figure 5), because gene replication plays an important role in the occurrence of novel functions and gene expansion. Analyses of homologous protein families are of great significance in establishing the kinship of species and predicting the function of new protein sequences [34,35]. On the basis of the phylogenetic relationships, the duplication events were proposed to occur in the *P. trichocarpa* genome. Nine *PtCAF1* genes (*PtCAF1I*/*PtCAF1M*, *PtCAF1K*/*PtCAF1M*, *PtCAF1M*/*PtCAF1P*, *PtCAF1I*/*PtCAF1K*, *PtCAF1I*/*PtCAF1P*, *PtCAF1F*/*PtCAF1L*, *PtCAF1F*/*PtCAF1Q*, *PtCAF1D*/*PtCAF1F*, *PtCAF1J*/*PtCAF1L*, *PtCAF1L*/*PtCAF1Q*, *PtCAF1D*/*PtCAF1K*, *PtCAF1J*/*PtCAF1Q*, *PtCAF1K*/*PtCAF1P*, and *PtCAF1D*/*PtCAF1Q*) were clustered into 14 repeat event regions on Chromosomes 1, 3, 4, 6, 9, 16, and 18 of *P. trichocarpa* (Appendix A). There are three clusters on Chromosome 6, indicating a hot spot of *PtCAF1* gene distribution. The results showed that part of the *PtCAF1* genes might be generated by gene replication, and the fragment repeat event was the main driving force of *PtCAF1* evolution. In order to further infer the phylogeny of *PtCAF1* family in *P. trichocarpa*, we constructed two comparative collinearity maps of poplar based on the developmental mechanism, including one dicotyledon (*A. thaliana*) and one monocotyledon (*O. sativa*). Four *PtCAF1* genes showed the same linear relationship with 2 *AtCAF1* genes in *A. thaliana*, and 3 *PtCAF1* genes showed the same linear relationship with one *OsCAF1* gene in *O. sativa*. Interestingly, the *PtCAF1L* and *PtCAF1Q* genes showed the same linear relationship in *A. thaliana*. It is suggested that these genes play an important role in the evolution of the *PtCAF1* gene family in *P. trichocarpa*.

In order to understand the evolutionary constraints of the *PtCAF1* family, we calculated the nonsynonymous to synonymous substitution ratios (Ka/Ks) of *PtCAF1* gene pairs (Table 2). All fragments and repeated *PtCAF1* gene pairs and most of the homologous *PtCAF1* genes had a Ka/Ks of < 1 except for *PtCAF1C* & *PtCAF1G*, which indicated that the *PtCAF1* gene family of poplar may have experienced a strong purification selection pressure during its evolution. The divergence time of the paralogous gene pairs of *PtCAF1F* & *PtCAF1L* and *PtCAF1C* & *PtCAF1G* were estimated to be about 200 million years ago, respectively. In brief, some *PtCAF1* genes might have been produced by gene replication, and these replication events were the main driving force of *PtCAF1* evolution.

### 2.7. Expression Patterns of PtCAF1 Genes in Different Plant Tissues

In order to investigate the function of the *PtCAF1* gene family in different tissues, the gene expression data of the 19 *PtCAF1* genes were obtained from the GEO datasets (NCBI GEO accession GSE81077). As shown in Figure 6, RNA-Seq and qRT-PCR data generated in the heatmap demonstrated a significant expression variation of *PtCAF1* in all vegetative and reproductive tissues (also see the analysis data in Appendix A). The transcriptional data showed that the expression of the *PtCAF1* gene in different tissues was different, and the expression of the *PtCAF1E*, *PtCAF1G*, *PtCAF1I*, *PtCAF1K*, *PtCAF1L*, *PtCAF1M*, *PtCAF1O*, *PtCAF1P*, *PtCAF1Q*, and *PtCAF1S* genes in the root was highest. Except for *PtCAF1K*, the expression of the aforementioned geens in the leaves was the lowest. The expression of the *PtCAF1A*, *PtCAF1B*, and *PtCAF1J* genes in the root was very low. The expression of *PtCAF1J* and *PtCAF1H* in the leaves was highest, and the expression of *PtCAF1A*, *PtCAF1B*, *PtCAF1C*, *PtCAF1F*, *PtCAF1N*, and *PtCAF1R* in the stem was also highest.

To further study the *CAF1* expression in different tissues of poplar, we studied the expression of the 19 *CAF1* genes in ‘Nanlin895′ poplar under normal growth conditions, and the transcription levels of the 19 *CAF1* genes in the roots, stems, young leaves, and mature leaves of ‘Nanlin895′ poplar were detected by quantitative reverse transcription PCR (qRT-PCR). Most *PtCAF1* genes were expressed in a wide range of tissues, and only two genes (*PtCAF1B* and *PtCAF1C*) could not be detected in the experiment. The qRT-PCR results showed that 17 *PtCAF1* genes were expressed in four tissues. The results showed that the expression levels of *PtCAF1E*, *PtCAF1G*, *PtCAF1I*, *PtCAF1J*, *PtCAF1K*, *PtCAF1L*, *PtCAF1M*, *PtCAF1O*, *PtCAF1P*, and *PtCAF1Q* were higher in the roots of ‘Nanlin895′ poplar, which was similar to the transcriptome data. The expression of *PtCAF1A*, *PtCAF1N*, and *PtCAF1R* in young leaves was higher than in mature leaves. The results showed that the expression of *PtCAF1* genes varied in different tissues, suggesting that the *PtCAF1* genes had multiple functions in poplar growth.

### 2.8. Expression Profiling of PtCAF1 in Response to Different Treatments

Poplar plays a significant role in the chemical industry in China. However, the growth of poplar is limited by environmental pressures such as salt, drought, and pathogenic bacteria [36,37]. In order to confirm whether the expression of the *PtCAF1* gene was affected by different stresses, 17 *PtCAF1* members were selected from 19 *PtCAF1* genes. We carried out various stress and hormone treatments on ‘Nanlin895′. The expression of the *PtCAF1* gene was detected by qRT-PCR under different conditions (see Figure 7 and the analysis data in Appendix A). The results showed that the expressions of 17 genes were up-regulated under ABA and salt treatment, among which *PtCAF1E*, *PtCAF1I*, *PtCAF1L*, *PtCAF1M*, *PtCAF1N*, and *PtCAF1Q* were significantly increased after ABA treatment for 12 h, and the PtCAF1 family after 6 h of salt treatment. In H2O2 treatment, most *PtCAF1* gene expression was up-regulated, and the expression of the *PtCAF1A*, *PtCAF1G*, *PtCAF1P*, *PtCAF1O*, and *PtCAF1R* genes was significantly increased after 12 h of treatment. In the PEG treatment, the expression of the *PtCAF1I*, *PtCAF1K*, *PtCAF1L*, *PtCAF1M*, and *PtCAF1Q* genes was increased considerably. The expression of the *PtCAF1A*, *PtCAF1O*, *PtCAF1P*, and *PtCAF1S* genes was significantly increased as a result of cold treatment. In the wound treatment, only the expression of the *PtCAF1F* gene was significantly increased. Interestingly, most genes reached the highest expression in 3 days of *Marssonina brunnea* treatment, and they were down-regulated at 5 days of *M. brunnea* treatment. At the same time, the changes in gene expression were not apparent in the process of 3 h to 6 h of JA and SA treatment; only *PtCAF1D*, *PtCAF1F*, *PtCAF1L*, and *PtCAF1Q* fluctuated. The results showed that some *PtCAF1* genes were induced or inhibited by multiple treatments. For example, the *PtCAF1D*, *PtCAF1F*, *PtCAF1J*, *PtCAF1L*, and *PtCAF1Q* genes in Group III were significantly up-regulated in all treatments and remained at a high level. In contrast, one treatment induced multiple *PtCAF1* genes at the same time. For example, 17 genes were significantly up-regulated in response to salt treatment.

## 3. Discussion

The *CAF1* gene is a relatively conservative gene that exists in a wide range of animals and plants. It has been extensively studied in yeast [38]. Thus far, the *CAF1* gene has been amply studied in animals, but there is a lack research in plants [39]. In this study, we analyzed the *PtCAF1* gene family in *P. trichocarpa* using phytozome v12.1. We identified 19 *PtCAF1* genes in *P. trichocarpa*. Moreover, we analyzed the structure and the chromosome location, performed a phylogenetic analysis, and checked the stress response mode of the *PtCAF1* gene family. These results lay a foundation for the further study of the *PtCAF1* gene. In the present study, 19 *PtCAF1* genes were identified, mainly distributed on eight chromosomes of *P. trichocarpa*. According to their chromosome location, they were named *PtCAF1A* to *PtCAF1S*. In order to explore the evolution of the *CAF1* gene family, the evolutionary tree of the *CAF1* gene was analyzed. We divided the poplar gene family into three group by distributing the *PtCAF1* gene family with *A. thaliana* and *O. sativa*. Most of the genes belong to Group II. Interestingly, by searching for Pham in the *A. thaliana* genome, it was found that, in addition to the 11 *AtCAF1* genes that were identified, two genes (*AtAHG2* and *AtRRD1*) have the *CAF1* domain. Furthermore, the results of the Pham search in the *O. sativa* genome are the same as those of previous studies. It was found that *AtAHG2* encodes a poly(A)-specific ribonuclease (*AtPARN*), which participates in the degradation of mRNA in plants [40]. In addition, SA and ABA play a key role in the external stress response [41]. Various results showed that the mutant ahg2-1 had higher endogenous SA, ABA, and various stress reactions in *A. thaliana* [42]. Moreover, various studies indicated that *AtRRD1* could participate in cell division during early lateral root organogenesis [43]. *PtCAF1E*, *PtCAF1N*, *PtCAF1O*, *AtAHG2*, and *AtRRD1* were in the same class, and *PtCAF1E*, *PtCAF1N*, and *PtCAF1O* were up-regulated to different degrees in the late response to stress. Therefore, *PtCAF1E*, *PtCAF1N*, and *PtCAF1O* may play an essential role in stress resistance in poplar. In addition, the subcellular localization of the gene is directly related to its function. The main function of the *CAF1* gene is to participate in the degradation of mRNA poly A, and 13 genes were found in the cytoplasm in the subcellular localization results. Other genes were located in the nucleus and chloroplast.

The study of exons and introns is helpful to understand the structural and functional differences of genes. Their acquisition or loss may be caused by gene duplication between different chromosomes [44,45]. In this study, we compared the protein sequences of *CAF1* members in *P. trichocarpa* and *A. thaliana*, and analyzed their phylogeny and gene structure. It was found that the *CAF1* family belonging to the same group usually has a similar motif composition and exon/intron structure. This indicates that they may have similar functions. Furthermore, specific sequence motifs in each group can endow the *CAF1* protein with specific functions, and the differences in these characteristics in different groups indicate the diversity of *PtCAF1* gene functions. Interestingly, the number of motifs of certain genes in Group II was significantly lower than that of other genes. These genes may have changed dramatically in evolution. We also found that the 19 *PtCAF1* genes contained different exons and introns, which indicates that there was a certain diversity among these three group genes. Regarding the distribution of introns and exons of the *PtCAF1* gene and *AtCAF1* gene, most Group I and Group III genes have no introns, and they contain at most one intron. The number of introns and exons of the *PtCAF1* genes in Group II is quite different: they contain seven introns or eight exons at most. The great difference in gene structure between different species indicates that significant changes took place in the long-term evolution of the poplar genome; however, the similarity of the gene structure in the same branch is very high, which indicates that the genes in the same group are very conservative as regards evolution.

Cis-elements play a key role in plant stress responses. For example, ABA-responsive elements (ABRE) respond to ABA hormone treatment, drought, and salt stress. LTR is involved in the plant response to low temperature. The TCA element and CGTCA motif were correlated with the expression level of MeJA and SA, respectively [46,47]. *PtCAF1* contains a variety of defense and stress response-related elements, indicating that *PtCAF1* plays an important role in biotic and abiotic stress. Plants have complex signal transduction networks in response to various external reactions, and they synthesize small hormones, such as MeJA and ABA, as signal molecules for the plant defense response [48]. A great deal of cis-elements responding to environmental stress were found in the promoter regions of the 19 *PtCAF1* genes, such as ABRE (the abiotic acid response), MBS (deficit induction), ARE (anaerobic induction), and TC-rich repeats (defense and stress response). ABRE, as an ABA-response element, is involved in regulating the drought response of poplar [49]. *PtCAF1D*, *PtCAF1L*, and *PtCAF1P* contained more than two ABRE elements, which may indicate that *PtCAF1D*, *PtCAF1L*, and *PtCAF1P* were involved in the response to drought stress. Many LTR (low-temperature responses) were found in the promoter region of *PtCAF1O*, which indicated that LTR is involved in the low temperature response of poplar. The results showed that the *PtCAF1* gene might participate in the response of poplar to environmental stress through ABA and other signaling channels.

Indeed, previous studies have reported that tandem and fragment replication events are important factors in the expansion and evolution of gene families. The occurrence of repetitive events in the genome helps plants adapt to the changing environment and evolve more smoothly [50,51,52,53]. Gene doubling events mainly include tandem replication, fragment replication, and reverse transcription transposition, which are important ways of gene family expansion [54]. Most of the homologous *PtCAF1* gene pairs had a Ka/Ks of <1, indicating that the *PtCAF1* gene may have been under selection pressure during evolution. Gene replication also plays an important role in promoting the evolution of the genome. Repetitive genes provide templates for new genes, which have new functions. There were 14 repeat event regions on seven chromosomes of *P. trichocarpa*, which indicates that repeat events played a key role in the expansion of the *PtCAF1* gene. The fragment replication of *PtCAF1* in *P. trichocarpa* mainly occurred in Group I and Group III. The collinearity analysis of *P. trichocarpa* with *A. thaliana* and *O. sativa* showed that gene replication mainly occurred in Group III, and the genetic relationship between *P. trichocarpa* and *A. thaliana* might be closer. The degree of differential expression of duplicate gene pairs is positively correlated with the time of differentiation: the longer the duplication, the more obvious the differentiation of the expression [55,56,57]. In this study, the *PtCAF1F* & *PtCAF1L* divergence time is the longest, and there are also various differences in the degree of response to different environments in later stages.

We analyzed the *PtCAF1* gene expression in four different tissues of ‘Nanlin895′. The results showed that *PtCAF1* gene expression was different in the four tissues. Most of the *PtCAF1* genes were highly expressed in the roots, while a small number of *PtCAF1* genes were expressed in the stems. This is consistent with the transcriptional data of *P. trichocarpa*. However, there are also differences. This may be due to the difference of some genes between *P. trichocarpa* and ‘Nanlin895′. The high expression level of the *PtCAF1* gene in the roots may reflect that the *PtCAF1* family plays an important role in responses to abiotic and biotic stresses from underground. Interestingly, the expression levels of 17 genes were high in the roots and leaves, but low in the stems. It may be that *PtCAF1* participates in the response of poplar to environmental stress, which leads to its unique expression pattern.

The *PtCAF1* gene plays an important role in regulating plant development, and giving plants tolerance to abiotic stresses (including salt, drought, heat, cold, and wound) [58]. In view of the importance of poplar in abiotic stress and the key role of the *PtCAF1* gene in physiological processes and stress responses, gene-specific expression usually reflects its corresponding function [59,60]. In this paper, the expression of *PtCAF1* in different environments was analyzed by qRT-PCR. Seventeen *PtCAF1* genes were differentially expressed in different environments, indicating that the *PtCAF1* gene may be involved in the response to stresses. It has been reported that *PtCAF1* is responsive to biotic stress in *O. sativa*, *A. thaliana*, and *C. sinensis*. Moreover, we found that most *PtCAF1* genes were significantly up-regulated after *M. brunnea* treatment for 3 days, with only *PtCAF1I* and *PtCAF1M* demonstrating no significant change. The results showed that the *PtCAF1* gene played an important role in responses to biological stress in poplar. Various recent studies report that the plant *CAF1* gene is mainly involved in responses to biotic stress. There is a lack of research on its response to abiotic stress, though various studies report that *CAF1* plays an important role in responses to low temperature stress in *O. sativa* and responses to wound stress in *A. thaliana* and *O. sativa* [27,29,61]. In this study, 17 genes were up-regulated after salt stress, indicating that *PtCAF1* plays a key role in responses to salt stress. In PEG, H_2_O_2_, and low temperature treatment, most genes were up-regulated. In addition, ABA plays a leading role in the regulation of plant adaptation to abiotic stresses, because ABA triggers many gene responses in plants and helps plants respond to environmental stimuli such as cold, drought, and salt [62,63,64,65]. In our study, after 12 h of ABA treatment, 17 *PtCAF1* genes were up-regulated to varying degrees. It may be that the *PtCAF1* gene is involved in the ABA metabolic regulation pathway, which causes *PtCAF1* to respond to abiotic stresses. In this study, the expression of the *PtCAF1* gene under cold, PEG, *M. brunnea*, H_2_O_2_, and salt stress differed. In addition, different stress can activate the same genes in different signaling pathways, which may be due to the production of common signaling components such as ABA or calcium caused by different stress stimuli [66,67].

As a model woody plant, poplar has a different life history to *A. thaliana* and *O. sativa* and may be more complex in terms of development and its gene regulation network [68,69,70]. Drought stress, salt stress, oxidation, plant pathogenic bacteria, and other factors seriously damage the growth and development of poplar. There is a lot of evidence to suggest that *PtCAF1* regulates plant growth and certain physiological processes and plays an important role in the response of the biotic and abiotic stress-induced defense signaling pathways [5,23]. Certain recent studies demonstrate that miRNA, which is a single stranded small RNA encoded by the higher eukaryotic genome, is also involved in mRNA degradation and in regulating the expression of more than one third of genes. It is an important component of the gene expression regulatory network and has an impact on expression levels [71]. In mammals, miRNA and *CAF1* can accelerate the deadenylation of the poly(A) tail of the target mRNA [72]. Furthermore, miR2275 and *CAF1* were found to be involved in early meiosis in male sterile wheat [5,73]. Therefore, miRNA and *CAF1* may regulate the expression of stress resistance genes in poplar. *PtCAF1* can participate in multiple abiotic stresses, and interaction with other proteins also plays a crucial role. In terms of potential applications, the *PtCAF1* gene has potential value in stress resistance, and targeting these genes may improve abiotic and biotic stress responses. In conclusion, the study provides an analysis of the *PtCAF1* gene family in *P. trichocarpa*, which provides a basis for the study of the potential function of the *PtCAF1* gene in poplar. These analyses can help to screen candidate *PtCAF1* genes for functional identification and provide a theoretical basis for the genetic improvement of the agronomic characteristics and environmental resistance of poplar.

## 4. Materials and Methods

### 4.1. Identification and Characterization of the CAF1 Family in P. trichocarpa

The complete *P. trichocarpa* protein sequences were downloaded from the JGI (Phytozome v12.1, https://phytozome.jgi.doe.gov/pz/portal.html (accessed on 30 April 2021)). To identify *P. trichocarpa CAF1* candidates, the Hidden Markov model (HMM) file corresponding to the *CAF1* domain (PF04857) was downloaded from the Pfam database (http://pfam.xfam.org/ (accessed on 30 April 2021)). HMMER 3.0 was used to search the *CAF1* gene from the *P. trichocarpa* genome database. The default parameter was determined and the cutoff was 1.2× 10^−28^. We also used it as the query to search the *P. trichocarpa* protein sequence data. To avoid missing probable *CAF1* members, the *P. trichocarpa* genome was used as the query to perform a BLAST search in the *A. thaliana* and *O. sativa CAF1* gene sequences, with a cutoff E-value of ≤10^−10^. The output putative *CAF1* protein sequences were submitted to the Pfam protein family database, SMART (Simple Modular Architecture Research Tool, http://smart.embl-heidelberg.de/ (accessed on 30 April 2021)), and CDD (Conserved Domain Database, https://www.ncbi.nlm.nih.gov/Structure/bwrpsb/bwrpsb.cgi (accessed on 30 April 2021)) to confirm the conserved *CAF1* domain.

### 4.2. Sequence Analysis

All the *PtCAF1* gene sequences were submitted to ExPASy (http://web.expasy.org/protparam/(accessed on 30 April 2021)) to find the number of amino acids, molecular weights (MW), and theoretical isoelectric points (PI). The *PtCAF1* subcellular targeting sites were identified using WOLF PSORT (https://wolfpsort.hgc.jp/ (accessed on 30 April 2021)). The chromosomal locations and intron numbers of *PtCAF1* were acquired through the *P. trichocarpa* genomic database. The motifs of *PtCAF1* proteins sequences were identified by MEME (http:/meme.nbcr.net/meme/intro.html (accessed on 30 April 2021)). The position information of the *CAF1* domains was obtained from the Pfam database, and the *PtCAF1* structure information was parsed from the General Feature Format files of *P. trichocarpa* using Perl script. Then, the exon and intron of the *PtCAF1* genes were identified on GSDS (the Gene Structure Display Server, http://gsds.cbi.pku.edu.cn/ (accessed on 30 April 2021)). In order to identify the cis-acting regulatory elements in promoters of the *PtCAF1* genes, 1500 bp upstream regions of the CDS were used to search the Plant Cis-acting Regulatory Elements (PlantCARE) database (http://bioinformatics.psb.ugent.be/webtools/plantcare/html/ (accessed on 30 April 2021)).

### 4.3. Multiple Sequence Alignment and Phylogenetic Analysis of CAF1 Proteins

Multiple sequence alignment was conducted using the Clustal X2.1 software with default settings. The *CAF1* protein sequences of *A. thaliana*, *O. sativa*, *C. annuum*, *C. sinensis*, *N. tabacum*, yeast (*SpCAF1*), human (HsCNOT7), mouse, *P. trichocarpa,* and *Z. mays* (*ZmCAF1*) were used for multiple sequence alignment. A phylogenetic tree was constructed in the Molecular Evolutionary Genetics Analysis (MEGA7) software (https://www.megasoftware.net/ (accessed on 30 April 2021)). According to the *CAF1* conserved domain, *PtCAF1* genes were divided into different groups. The *CAF1* protein sequences for *O. sativa*, *A. thaliana*, *E. grandis*, *E. guineensis*, *C. annuum*, *C. sinensis*, *P. trichocarpa*, and *N. tabacum* were used for the phylogenetic tree analysis. The *CAF1* protein sequences of *A. thaliana*, *E. grandis* [74], and *O. sativa* were downloaded from the JGI gene catalog (Phytozome v12.1, https://phytozome.jgi.doe.gov/pz/portal.html (accessed on 30 April 2021)). The *C. annuum* (*CaCAF1*), *C. sinensis*, yeast (*SpCAF1*), mouse, human, *E. guineensis*, and *Z. mays CAF1* protein sequences were downloaded from the National Center for Biotechnology Information (https://www.ncbi.nlm.nih.gov/ (accessed on 30 April 2021)). All sequences of different species’ *CAF1* genes are listed in Appendix A. Moreover, all species’ *CAF1* protein sequences are listed in Appendix A. The phylogenetic trees were constructed using the neighbor-joining (NJ) method, the gaps/missing data treatment method was used for partial deletion, the specific model/method was the Poisson model, the number of bootstrap replications was 1000, and the site coverage cutoff was 0.95 [75].

### 4.4. Prediction of Protein Spatial Structure

In order to predict the spatial structure of proteins, the SWISS-MODEL website (https://swissmodel.expasy.org/ (accessed on 30 April 2021)) was used to analyze the *CAF1* protein structures of *P. trichocarpa*, *O. sativa*, and *A. thaliana*. The protein spatial structure was constructed using the UCSF Chimera software (https://www.cgl.ucsf.edu/chimera/ (accessed on 30 April 2021)).

### 4.5. Chromosomal Locations, Gene Duplication of PtCAF1 Genes, and Calculation of Ka/Ks

The chromosomal positions of the *PtCAF1* genes were obtained from the *P. trichocarpa* genome. MapChart was used for the mapping of *PtCAF1* chromosomal positions and distances. In order to understand the evolutionary constraints of *PtCAF1*, we calculated the Ka/Ks ratio of the *PtCAF1* gene pairs. Hits with an E-value of <10^−5^ and a homology greater than 80% were considered significant. For estimating the Ks and Ka substitution rates, the amino acids of paralogous and orthologous *PtCAF1* proteins were analyzed using the Ka/Ks calculator. The divergence time was estimated with the formula T = Ks/2r. The r was taken to be 1.5 × 10^−8^ synonymous substitutions per site per year for dicotyledonous plants [76]. For the homology analysis, we studied the repetitive events of *PtCAF1* genes in the *A. thaliana* and *O. sativa* genomes from the plant genome duplication database. The analysis of gene replication events was conducted with multiple collinear scanning toolkits (MCScanX). Then, the synteny blocks were illustrated with the CIRCOS software (http://circos.ca/software/download/circos (accessed on 30 April 2021)) between *P. trichocarpa* genes in *A. thaliana* and *O. sativa*.

### 4.6. Transcriptional Analyses

The *P. trichocarpa* cv. and *P. deltoides × P. euramericana* cv. ‘Nanlin895′ were used in this study. *P. trichocarpa* and ‘Nanlin 895′ were cultured in an artificial climate chamber with a 16 h/8 h photoperiod in 1/2 Murashige and Skoog (MS) medium (pH 5.8) at a temperature of 23 ± 1 °C. RNA was extracted from young and mature leaves, stems, and roots using a Plant RNA Extraction Kit (Biomiga, Inc., San Diego, CA, USA). *M. brunnea* was cultured in potato medium (PDA) at 25 °C. WT poplars grown for 3 months were treated with 200 mM ABA, 200 mM SA, 200 m JA, 2 mM H_2_O_2_, and 200 mM NaCl, and samples were taken at 0, 3, 6, and 12 h. They were also treated with Macrogol 6000 (PEG6000), *M. brunnea*, at 4 °C (cold stress), and sampled at 0, 1, 3, and 5 days. We also wounded leaves at 23 °C, and samples were taken at 0, 3, 6, and 12 h. RNA was then extracted from the leaves using a Plant RNA Extraction Kit (Biomiga, Inc.).

Premier 5 was used to design primers specific to the *PtCAF1* genes (Appendix A). Total RNA was extracted using a Plant RNA Extraction Kit (Biomiga, Inc.). The cDNA was obtained using a First Strand cDNA Synthesis Kit (TaKaRa, Shiga, Japan). First-strand cDNA in the reactions comprised a 500 ng RNA template, and all the procedures were carried out according to the instructions provided by the kit. Before the qRT-PCR analysis, the template was diluted 10 times. The relative expression level of the gene was calculated by comparing the average of *EF1**α* (Potri.006G130900.1) and *PtActin* (Potri.019G006700.1) internal reference genes. The qRT-PCR reaction system included 10 μL SYBR Green, 1 μL forward primer, 1 μL reverse primer, 1 μL cDNA, and 7 μL ddH_2_0. qRT-PCR involved 40 cycles of pre-denaturation at 98 °C for 10 min, denaturation at 95 °C for 10 s, and annealing at 60 °C for 30 s. The method of 2^−ΔΔCT^ was used to analyze the data in this paper [77].

## 5. Conclusions

In this study, we identified 19 poplar *PtCAF1* genes. The classification, gene structure, and evolutionary characteristics of the *PtCAF1* genes in poplar showed that *CAF1* was relatively conservative in terms of the evolutionary process. The differential expression of the *PtCAF1* genes in poplar tissues showed that they played different roles in the development of poplar, and many genes showed tissue-specific expression patterns. In addition, a *PtCAF1* gene expression analysis showed that certain genes were significantly up-regulated or down-regulated under biotic and abiotic stresses. Our results also revealed the differences in the expression of *PtCAF1* induced by biotic and abiotic stresses in poplar, indicating that these *PtCAF1* genes are involved in abiotic stress tolerance. In conclusion, our study established the structural and functional framework of the *PtCAF1* protein. Although poplar genome sequencing has been carried out for many years, the identification and function of poplar stress-related genes still remains to be studied. Our results will be helpful for the further study of the important role of the *PtCAF1* gene in poplar stress responses. It may also be utilized in molecular breeding programs for poplar stress tolerance.

## Figures and Tables

**Figure 1 plants-10-00981-f001:**
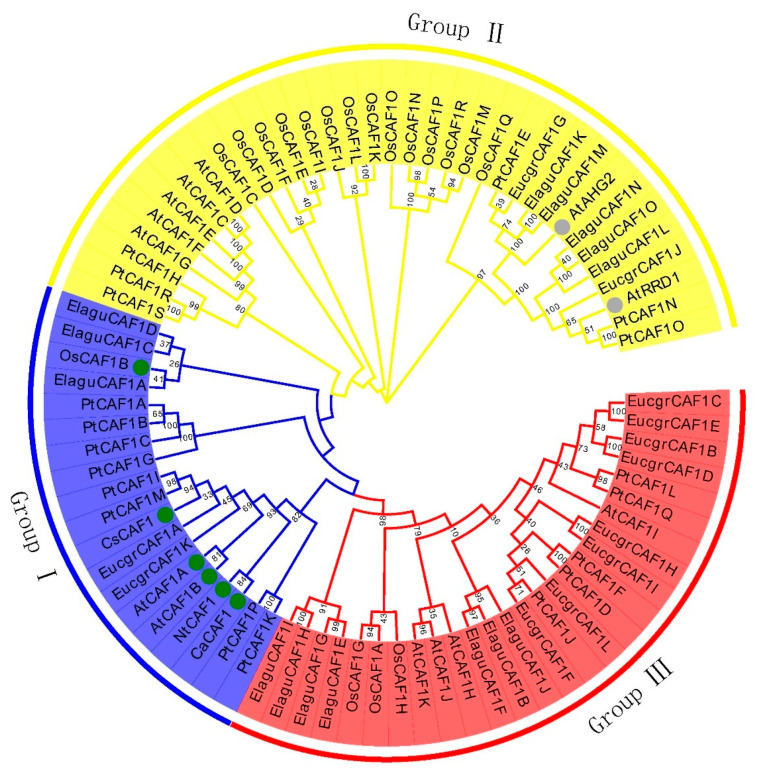
Phylogenetic relationship between *P. trichocarpa* and other plant species. Phylogenetic analyses of the *PtCAF1* protein alignments of *CAF1* domains among *Oryza sativa*, *Arabidopsis thaliana*, *Eucalyptus grandis*, *Elaeis guineensis*, *Capsicum annuum*, *Citrus sinensis*, and *Nicotiana tabacum* were performed with ClustalX2.1, and the phylogenetic tree was constructed using the NJ method with MEGA 7.0. The circle represents the gene whose function has been studied. The blue color denotes Group I, the yellow color denotes Group II, and the red color denotes Group III.

**Figure 2 plants-10-00981-f002:**
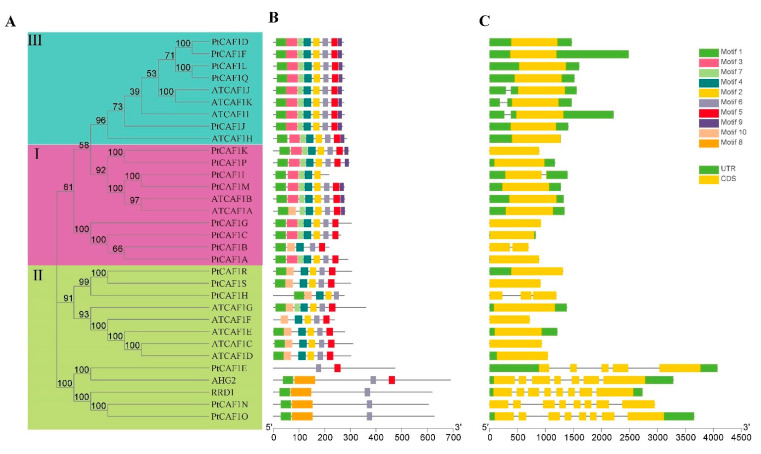
Phylogenetic relationships, gene structure, and architecture of conserved protein motifs in *CAF1* from *P. trichocarpa* and *A. thaliana*. (UTR:Untranslated Regions. CDS: Coding DNA Sequence.) (**A**) The phylogenetic tree was constructed based on the full-length sequences of *P. trichocarpa* and *A. thaliana* CAF1 proteins using MEGA 7.0 software. (**B**) The motif composition of *P. trichocarpa* and *A. thaliana CAF1* proteins. The motifs, numbers 1–10, are displayed in different colored boxes. The protein length can be estimated using the scale at the bottom. (**C**) Exon/intron structure of *P. trichocarpa* and *A. thaliana CAF1* genes. Green boxes indicate untranslated 5′ and 3′ regions, yellow boxes indicate exons, and black lines indicate introns.

**Figure 3 plants-10-00981-f003:**
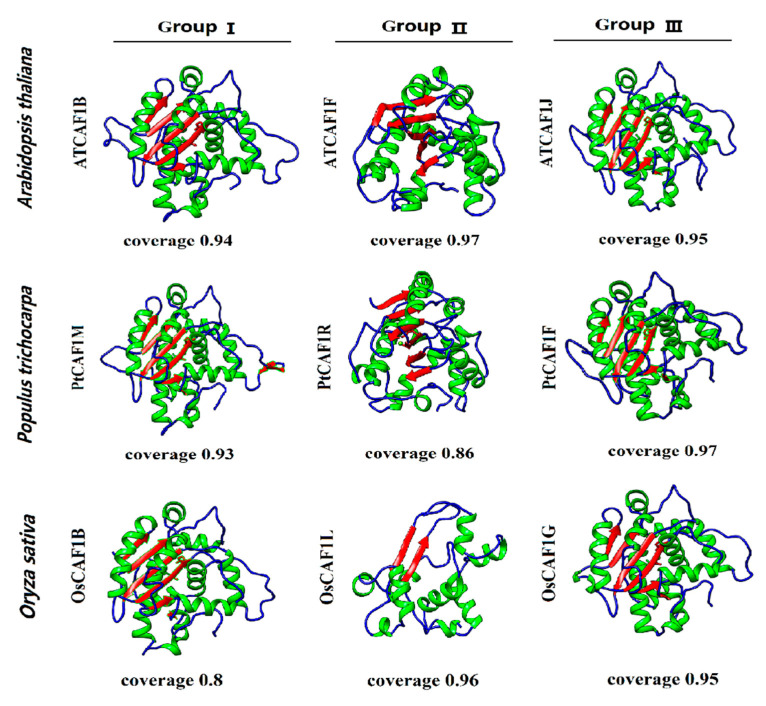
Prediction of the spatial structure of the *CAF1* protein. The protein coil is blue, the helix is green, and the strand is red.

**Figure 4 plants-10-00981-f004:**
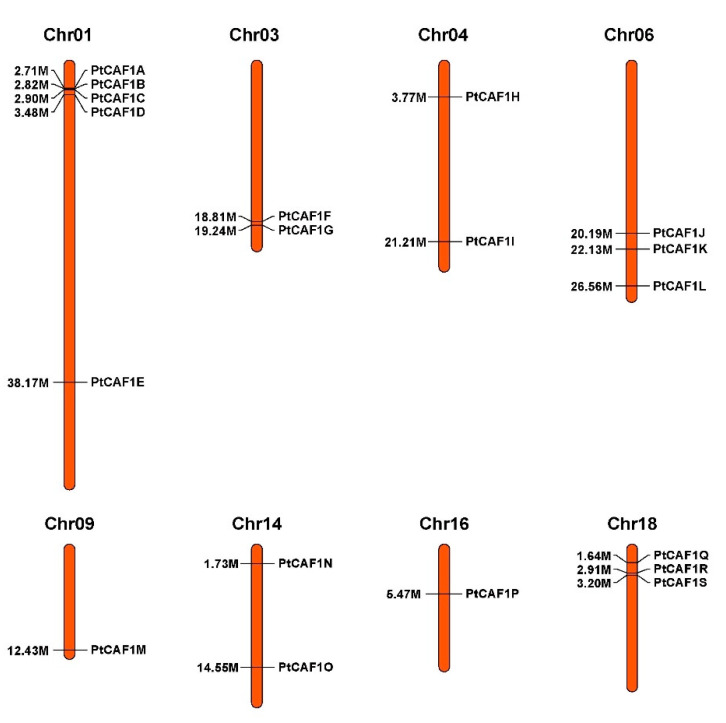
Chromosomal distribution of the 19 *PtCAF1* genes. The scale of the chromosome is in megabases (Mb). The chromosome number is indicated at the top of each chromosome.

**Figure 5 plants-10-00981-f005:**
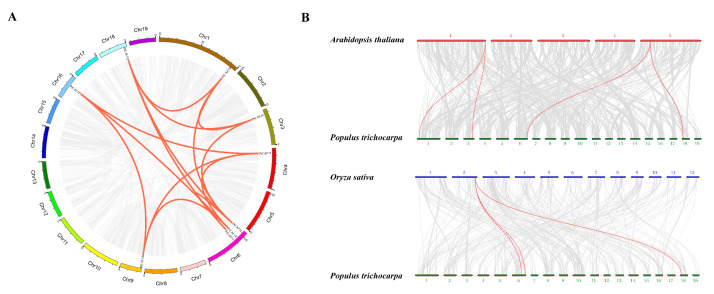
Duplication event analysis of *CAF1* genes and comparative synteny analysis among *P. trichocarpa*, *A. thaliana,* and *O. sativa*. Gray lines in the background indicate the collinear blocks within *P. trichocarpa* and other plant genomes, while red lines highlight syntenic *PtCAF1* gene pairs. (**A**) The data were derived from the Plant Genome Duplication Database, and 14 couples of duplicated *PtCAF1* genes were anchored to corresponding positions on *P. trichocarpa* chromosomes using the CIRCOS program. (**B**) Synteny between *P. trichocarpa* and *A. thaliana* or between *P. trichocarpa* and *O. sativa* was anchored to the corresponding position on specific chromosomes using the CIRCOS program. *P. trichocarpa* chromosomes are depicted as green segments, and *A. thaliana* and *O. sativa* are shown in red and blue, respectively. The size of chromosomes was consistent with the actual pseudo-chromosome size.

**Figure 6 plants-10-00981-f006:**
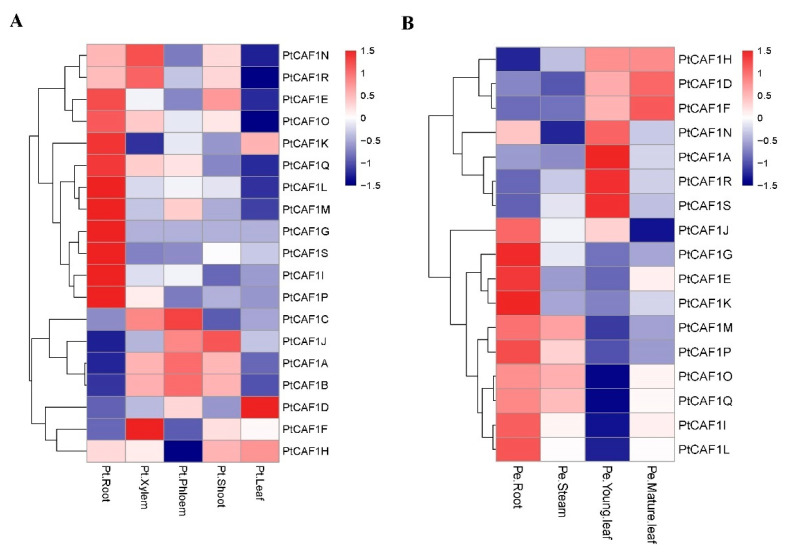
Tissue-specific gene expression of *PtCAF1* genes. The expression patterns of *PtCAF1* genes in leaf, root, and stem tissues were examined by RNA-Seq and qRT-PCR data. The heatmap was created based on the log2-transformed values of the relative expression levels of the *PtCAF1* genes. (**A**) *PtCAF1* gene family expression in different *P. trichocarpa* tissues and organs. (**B**) Tissue-specific expression of *PtCAF1* in ‘Nanlin 895′. Expression levels are represented in various colors, with red indicating higher expression levels and navy indicating lower expression levels.

**Figure 7 plants-10-00981-f007:**
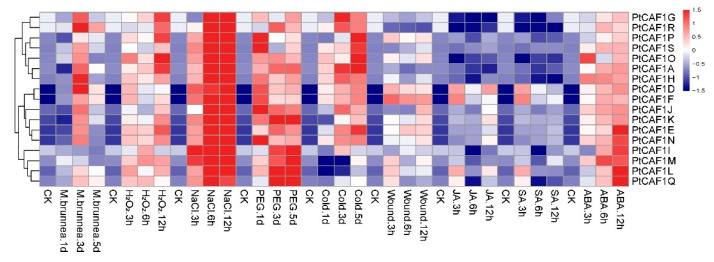
The expression profiles of the *PtCAF1* gene family in poplar under abiotic or biotic stresses and phytohormone treatments. The heatmaps were constructed using the R package based on the expression datasets. Expression levels are represented in various colors, with red indicating higher expression levels and navy indicating lower expression levels.

**Table 1 plants-10-00981-t001:** List of 19 *PtCAF1* genes and their basic characterizations.

Name	Phytozome Gene ID	Ensembl Gene ID	CDS	AA	MW (kDa)	pI	Subcellular Location
*PtCAF1A*	Potri.001G038500.1	PNT52564	876	292	33.40	6.86	cytosol
*PtCAF1B*	Potri.001G039000.1	PNT52569	648	216	24.78	8.47	cytosol
*PtCAF1C*	Potri.001G040400.1	PNT52586	789	263	30.25	9.91	cytosol
*PtCAF1D*	Potri.001G046700.1	PNT52695	825	275	31.30	4.56	cytosol
*PtCAF1E*	Potri.001G368400.1	PNT58721	1425	475	53.50	7.85	nucleus
*PtCAF1F*	Potri.003G181100.1	PNT46250	825	275	31.22	4.49	cytosol
*PtCAF1G*	Potri.003G186300.1	PNT46340	915	305	34.53	5.21	nucleus
*PtCAF1H*	Potri.004G048800.1	PNT39610	828	276	31.98	6.37	chloroplast
*PtCAF1I*	Potri.004G200400.1	PNT42179	651	217	25.12	4.62	cytosol
*PtCAF1J*	Potri.006G187200.1	PNT32402	810	270	30.31	5.34	cytosol
*PtCAF1K*	Potri.006G205600.1	PNT32745	888	296	33.70	5.96	chloroplast
*PtCAF1L*	Potri.006G262500.1	PNT33885	834	278	31.49	4.46	cytosol
*PtCAF1M*	Potri.009G161500.1	PNT21675	834	278	31.71	5.08	cytosol
*PtCAF1N*	Potri.014G018500.1	PNT02530	1815	605	67.72	7.27	chloroplast
*PtCAF1O*	Potri.014G177400.1	PNT05527	1881	627	70.23	6.12	chloroplast
*PtCAF1P*	Potri.016G073000.1	PNS98364	891	297	33.43	5.8	cytosol
*PtCAF1Q*	Potri.018G020900.1	PNS92197	834	278	31.42	4.41	cytosol
*PtCAF1R*	Potri.018G036100.1	PNS92495	918	306	33.97	9.1	cytosol
*PtCAF1S*	Potri.018G038700.1	PNS92540	909	303	33.75	8.8	cytosol

Note: CDS is coding sequence length, AA is protein amino acid quantity, MW is protein molecular weight, PI is isoelectric point.

**Table 2 plants-10-00981-t002:** Ka/Ks analysis of the *PtCAF1* gene pairs duplication.

Sequence	Ka	Ks	Ka/Ks	*p*-Value (Fisher)	Length	Time (Mya)
*PtCAF1D* & *PtCAF1F*	0.016708	0.306409	0.05453	1.36 × 10^−22^	822	102.1363
*PtCAF1A* & *PtCAF1C*	0.014891	0.019054	0.781482	0.429132	780	6.351433
*PtCAF1C* & *PtCAF1G*	1.01436	0.952687	1.06473	0.247271	723	317.5623
*PtCAF1A* & *PtCAF1G*	0.12254	0.316388	0.387308	5.67 × 10^−8^	873	105.4627
*PtCAF1F* & *PtCAF1L*	0.082084	2.14757	0.038222	4.1× 10^−125^	819	715.8567
*PtCAF1K* & *PtCAF1P*	0.049893	0.434185	0.114912	8.21× 10^−28^	858	144.7283
*PtCAF1N* & *PtCAF1O*	0.008309	0.01364	0.609174	0.239107	1797	4.546667
*PtCAF1L* & *PtCAF1Q*	0.034304	0.337114	0.101757	2.86× 10^−20^	831	112.3713
*PtCAF1R* & *PtCAF1S*	0.028137	0.084529	0.332862	0.000561	906	28.17643

Note: Ka: The ratio of the number of nonsynonymous substitutions per nonsynonymous site. Ks: the number of synonymous substitutions per synonymous site. Ka/Ks: the nonsynonymous to synonymous substitution ratios. Mya: million years ago.

## Data Availability

Not applicable.

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
