# Peer review of "Genome-Wide and Comprehensive Analysis of the Multiple Stress-Related CAF1 (CCR4-Associated Factor 1) Family and Its Expression in Poplar"

_plants, 2021, doi:10.3390/plants10050981_

Round 1

Reviewer 1 Report

The presented work by Wang et al., represents an in silico analysis of PtCAF1 gene family in Poplar (Populus spp.). They identified 19 potential PtCAF1 genes in P. trichocarpa genome, illustrated phylogenetic relationship between P. trichocarpa and other species, summarized gene structure and protein predicted form, conserved motifs of CAF1 group in this species. They performed the promoter analysis to list out the potential cis-regulatory elements, chromosomal distribution of those PtCAF1 genes, duplication event analysis and comparative synteny among P. trichocarpa, A. thaliana and O. sativa. Authors also gathered gene expression data from NCBI GEO accession GSE81077 for PtCAF1 genes across different tissues and organs of P. trichocarpa. Additionally, they did qRT-PCR to assess the expression of 19 PtCAF1 genes in ‘Nanlin895’ poplar under normal growth conditions and under abiotic or biotic stresses and phytohormone treatments. Authors also claimed that this exhaustive analysis would help the Poplar molecular breeding to get biotechnological benefits from findings.

However, I have certain concerns regarding how data is represented. Addressing these issues might help author improving the current form of Manuscript. Major concern is associated with the figure quality.

Methods:

Section 2.2, authors should indicate all the parameters they used to build the phylogeny tree using NJ method even it is the default of software: pairwise deletion, Poisson model or else, etc.

Line 186, the reference of SWISS model should be added.

Line 552, the reference of UCSF Chimera software should be added.

A minor point, MEME version 7.0 is old one. Even it might not affect the output, it is better to use the most updated version.

qRT-PCR part: author should provide more information about the significant test that was performed to assess the gene expression difference.

Results:

Figure 1: The letter size of bootstrap value is quite small. The legend of figure should include the description of each group.

Figure 4: Color codes of the cis-elements are quite hard to differentiate. This figure can be transferred to supplementary data.

Line 288: Why only 2 genes PtCAF1B and PtCAF1C could not be detected? Could authors provide some information about these problems?

Figure 7B: Is there any significant test that was performed to assess the gene expression difference? When authors mentioned “higher” (line 292), does it mean significantly higher?

Author Response

Dear Editor:

Thank you for your consideration for publication of our manuscript, Genome-wide identification and expression analysis of the multiple stress-related CAF1 (CCR4 assoicated factor 1) family in poplar, which submitted to plants.

Thanks for the valuable suggestions and comments to our last submitted manuscript. We modified the manuscript according to the reviewer’s comments, and the following is the responses to the reviewer’s comments.

Point 1: Section 2.2, authors should indicate all the parameters they used to build the phylogeny tree using NJ method even it is the default of software: pairwise deletion, Poisson model or else, etc.

Response 1: Thanks for your comment. It has been modified accordingly.

Point 2: Line 186, the reference of SWISS model should be added.

Response 2: Thanks for your comment. It has been modified accordingly.

Point 3: Line 552, the reference of UCSF Chimera software should be added.

Response 3: Thanks for your comment. It has been modified accordingly.

Point 4: A minor point, MEME version 7.0 is old one. Even it might not affect the output, it is better to use the most updated version.

Response 4: Thanks for your comment. It has been modified accordingly.

Point 5: qRT-PCR part: author should provide more information about the significant test that was performed to assess the gene expression difference.

Response 5: Thanks for your comment. We use different colors to represent the relative expression of different genes, red color indicating higher expression levels and navy color indicating lower expression levels. And we provide some quantitative data in the supplementary materials.

Point 6: Figure 1: The letter size of bootstrap value is quite small. The legend of figure should include the description of each group.

Response 6: Thanks for your comment. It has been modified accordingly.

Point 7: Figure 4: Color codes of the cis-elements are quite hard to differentiate. This figure can be transferred to supplementary data.

Response 7: Thanks for your comment. It has been modified accordingly.

Point 8: Line 288: Why only 2 genes PtCAF1B and PtCAF1C could not be detected? Could authors provide some information about these problems?

Response 8: Thanks for your comment. The PtCAF1B and PtCAF1C were difficult to detect these two genes by qRT-PCR because of their very low expression.

Point 9: Figure 7B: Is there any significant test that was performed to assess the gene expression difference? When authors mentioned “higher” (line 292), does it mean significantly higher?

Response 9: Thanks for your comment. It has been modified accordingly. We use different colors to represent the relative expression of different genes, red indicating higher expression levels and navy indicating lower expression levels. And we provide some quantitative data in the supplementary materials.

Thank you for reviewer’s advice, and the manuscript was revised accordingly. All the changes were highlighted in the revised manuscript.

Thank you for considering our manuscript for publication.

Sincerely yours,

Qiang Zhuge, Professor

Nanjing Forestry University,

Nanjing, Jiangsu, China, 210037

Reviewer 2 Report

In the paper “Genome-wide identification and expression analysis of the multiple stress-related CAF1 (CCR4 associated factor 1) family in poplar”, a comprehensive analysis of the gene structure, protein sequence and phylogeny of the CAF1 family, involved in stress response regulation, in poplar has been performed. In addition, the expression of 19 PtCAF1 genes was evaluated in different tissues and under biotic and abiotic stresses. The study underlines the importance of the PtCAF1 gene family in regulating poplar stress response and provides useful bases for breeding programs aimed at ameliorating poplar stress tolerance. It represents a well conducted research that deserves to be published after improvements.

The reviewer suggests the present paper to be included in the Special issue "Poplar Responses to Environmental Stresses" ISSN 2223-7747.

Please consider the following comments.

  • The title can be changed in “Genome-wide and expression comprehensive analysis of the multiple stress-related CAF1 (CCR4 associated factor 1) family in poplar”.
  • The first sentence in Introduction should be modified. Other factors (i.e., epigenetics), not only gene expression, contribute to plant/environmental interaction.
  • References must be improved across the whole manuscript. For example, lines 55 and 60 need citations.
  • A list of the motifs, with the corresponding sequence and putative function, identified in the protein sequences should be provided. The main motifs can be also highlighted in the protein structures (Fig. 3).
  • Please provide an accurate description of the experimental design of qRT-PCRs in the Results section. A table might help.
  • The Discussion section contains parts that can be moved to Results.
  • In lines 418 and 419 authors declared that, together with tandem replication and fragment replication, reverse transcription and transposition participate to gene doubling. Did the authors check for the presence of retroelements in the PtCAF1 gene clusters?
  • From lines 485 to 489, authors indicated that CAF1 regulates mRNA degradation with the action of miRNAs in different species. Did the authors check if such miRNAs are present also in poplar genome?
  • Paragraphs 4.6 and 4.7 should be merged in a unique 4.6 paragraph entitled, for example, “transcriptional analyses”.
  • The resolution of the figures must be increased.
  • Supplementary Table 5 should be moved to the main text.
  • Please check all the figure captions. For example, in the caption of Figure 2 UTRs are reported as yellow boxes but they are shown as green boxes in the figure. Moreover, in lines 275 and 276, authors declared that Figure 7 shows RNA-Seq data, however, there is no indication of RNA-Seq in the caption; only qRT-PCRs are cited.
  • Please check all the Supplementary files. In the main text, Supplementary Table S3 is cited before Supplementary Tables S1 and S2. Please add headers to each column in Supplementary Tables S2, S3 and S4. Supplementary Table S2 is written in Chinese.
  • English must be extensively improved.

Author Response

Dear Editor:

Thank you for your consideration for publication of our manuscript, Genome-wide identification and expression analysis of the multiple stress-related CAF1 (CCR4 assoicated factor 1) family in poplar, which submitted to plants.

Thanks for the valuable suggestions and comments to our last submitted manuscript. We modified the manuscript according to the reviewer’s comments, and the following is the responses to the reviewer’s comments.

Point 1: The title can be changed in “Genome-wide and expression comprehensive analysis of the multiple stress-related CAF1 (CCR4 associated factor 1) family in poplar”.

Response 1: Thanks for your comment. It has been modified accordingly.

Point 2: The first sentence in Introduction should be modified. Other factors (i.e., epigenetics), not only gene expression, contribute to plant/environmental interaction.

Response 2: Thanks for your comment. It has been modified accordingly.

Point 3: References must be improved across the whole manuscript. For example, lines 55 and 60 need citations.

Response 3: Thanks for your comment. It has been modified accordingly.

Point 4: A list of the motifs, with the corresponding sequence and putative function, identified in the protein sequences should be provided. The main motifs can be also highlighted in the protein structures (Fig. 3).

Response 4: Thanks for your comment. The list of the motifs was provided in supplementary figure 2.

Point 5: Please provide an accurate description of the experimental design of qRT-PCRs in the Results section. A table might help.

Response 5: Thanks for your comment. It has been modified accordingly.

Point 6:  The Discussion section contains parts that can be moved to Results.

Response 6: Thanks for your comment. It has been modified accordingly.

Point 7: In lines 418 and 419 authors declared that, together with tandem replication and fragment replication, reverse transcription and transposition participate to gene doubling. Did the authors check for the presence of retroelements in the PtCAF1 gene clusters?

Response 7: Thanks for your comment. All of cis-elements are in supplement table 1, and we check again carefully. There was no reverse transcriptase element in ptcaf1 gene cluster.

Point 9: From lines 485 to 489, authors indicated that CAF1 regulates mRNA degradation with the action of miRNAs in different species. Did the authors check if such miRNAs are present also in poplar genome?

Response 8: Thanks for your comment. We check some miRNAs through some websites (http://www.mirbase.org/) (http://plantgrn.noble.org/v1_psRNATarget/), such as miR2275, MiR1122, miR1127a, but there is no report about the target of miRNA in poplar.

Point 10: Paragraphs 4.6 and 4.7 should be merged in a unique 4.6 paragraph entitled, for example, “transcriptional analyses”.

Response 10: Thanks for your comment. It has been modified accordingly.

Point 11: The resolution of the figures must be increased.

Response 11: Thanks for your comment. It has been modified accordingly.

Point 12: Supplementary Table 5 should be moved to the main text.

Response 12: Thanks for your comment. It has been modified accordingly.

Point 13: Please check all the figure captions. For example, in the caption of Figure 2 UTRs are reported as yellow boxes but they are shown as green boxes in the figure. Moreover, in lines 275 and 276, authors declared that Figure 7 shows RNA-Seq data, however, there is no indication of RNA-Seq in the caption; only qRT-PCRs are cited.

Response 13: Thanks for your comment. It has been modified accordingly.

Point 14: Please check all the Supplementary files. In the main text, Supplementary Table S3 is cited before Supplementary Tables S1 and S2. Please add headers to each column in Supplementary Tables S2, S3 and S4. Supplementary Table S2 is written in Chinese.

Response 14: Thanks for your comment. It has been modified accordingly.

Thank you for reviewer’s advice, and the manuscript was revised accordingly. All the changes were highlighted in the revised manuscript.

Thank you for considering our manuscript for publication.

Sincerely yours,

Qiang Zhuge, Professor

Nanjing Forestry University,

Nanjing, Jiangsu, China, 210037

Round 2

Reviewer 1 Report

I think authors have sufficiently reponded to concerns raised. 

Author Response

Dear professor,

Thanks for your careful review  and valuable suggestions. 

Reviewer 2 Report

Dear authors,

the reviewer appreciates all the changes in the manuscript. Nonethless some English mistakes still occur. Please check.

The reviewer suggests to implement the caption of Supplementary Figure 2.

Did the authors consider to submit the manuscript on the Special issue ISSN: 2223-7747 "Poplar Responses to Environmental Stresses"?

Author Response

Dear professor:

Thanks for your careful review and valuable suggestions.  All have  been modified accordingly.

Point 1: the reviewer appreciates all the changes in the manuscript. Nonethless some English mistakes still occur. Please check

Response 1: Thanks for your valuable comment. We have cheched the manuscript again carefully and did language modification. The modification has been completed accordingly.

.Point 2: The reviewer suggests to implement the caption of Supplementary Figure 2.

Response 2: Thanks for your comment. It has been modified accordingly.

Point 3: Did the authors consider to submit the manuscript on the Special issue ISSN: 2223-7747 "Poplar Responses to Environmental Stresses"?

Response 3: Thanks for your comment. We agree to submit the manuscript on the Special issue ISSN: 2223-7747 "Poplar Responses to Environmental Stresses” if the editor thinks it is necessary.

Round 3

Reviewer 2 Report

The reviewer appreciates the corrections made by authors, nonetheless many grammer mistakes still occur.

For example:

Line 35 mRNA should be explained at the first appearance.

Sentence in line 54 sounds weird.

Article is not needed in lines 58 and 59.

Line 110 is grammatically wrong.

The reviewer strongly recommend to check the manuscript again. The revision by a native English speaker could help.

Sincerely

Author Response

Dear Reviewer:

Thanks for the valuable suggestions and comments to our last submitted manuscript. We modified the manuscript according to the reviewer’s comments, and the following is the responses to the reviewer’s comments. The paper has been revised at https://www.mdpi.com/authors/english.